# Methodology for the Differential Classification of Dengue and Chikungunya According to the PAHO 2022 Diagnostic Guide

**DOI:** 10.3390/v16071088

**Published:** 2024-07-06

**Authors:** Wilson Arrubla-Hoyos, Jorge Gómez Gómez, Emiro De-La-Hoz-Franco

**Affiliations:** 1Facultad de Ingeniería, Universidad Nacional Abierta ya Distancia, Sincelejo 700002, Colombia; wilson.arrubla@unad.edu.co; 2Grupo SOCRATES, Departamento de Ingeniería de Sistemas y Telecomunicaciones, Facultad de Ingeniería, Universidad de Córdoba, Montería 230001, Colombia; 3Department of Computer Science and Electronics, Faculty of Engineering, Universidad de la Costa, Barranquilla 080002, Colombia; edelahoz@cuc.edu.co

**Keywords:** PAHO, dengue, Zika, chikungunya, linear interpolation, machine learning, sets, medical evidence synthesis

## Abstract

Arboviruses such as dengue, Zika, and chikungunya present similar symptoms in the early stages, which complicates their differential and timely diagnosis. In 2022, the PAHO published a guide to address this challenge. This study proposes a methodological framework that transforms qualitative information into quantitative information, establishing differential weights in relation to symptoms according to the medical evidence and the GRADE scale based on recommendation 1 of the said guide. To achieve this, common variables from the dataset were identified using the PAHO guide, and quality rules were established. A linear interpolation function was then parameterised to assign weights to the symptoms according to the evidence. Machine learning was used to compare the different models, achieving 99% accuracy compared with 79% without the methodology. This proposal represents a significant advancement, allowing the direct application of the PAHO recommendations to the dataset and improving the differential classification of arboviruses.

## 1. Introduction

The General Assembly of the United Nations formulated the 2030 Agenda for Sustainable Development, which comprises a range of measures intended to safeguard both human well-being and global health. This plan is structured around 17 axes known as the Sustainable Development Goals (SDGs), with SDG 3—Health and Well-being committing to ending epidemics such as AIDS, tuberculosis, and malaria, as well as combating hepatitis and other neglected tropical diseases [1].

In 2022, significant increases were recorded in the cases of dengue, Zika, and chikungunya. These tropical diseases are arboviral infections transmitted by Aedes aegypti and Aedes albopictus [2,3], which circulate epidemics throughout the Americas region. Of the total cases, 2,811,433 (90%) were dengue, 273,685 (8.7%) were chikungunya, and 40,249 (1.3%) were Zika [4]. In 2023, outbreaks of chikungunya and dengue surpassed the expected cases in South America, Central America, and the Caribbean. Between weeks 1 and 23, 2,216,405 cases were recorded, of which 1,994,088 (90%) were dengue, 213,561 (9.6%) were chikungunya, and 8756 were Zika [4].

These viral diseases pose a constant threat to global public health. According to the records from the World Health Organization (WHO), only dengue has multiplied by 10 in reported cases, increasing from 500,000 to 5.2 million in 129 countries, with 80% of reported cases in the American region [5]. It is the most severe condition, as it can cause dengue shock syndrome owing to plasma loss, leading to death [6]. The early diagnosis of these diseases poses a challenge, as they share a similar clinical picture [7], especially when co-circulation occurs in endemic areas [8].

To address these challenges, in 2022, the Pan American Health Organization (PAHO) proposed guidelines for the diagnosis and treatment of dengue, Zika, and chikungunya in the American region. These guidelines were developed by experts in the field following a systematic literature review and discussion. This guide summarises 12 recommendations applicable to adult and paediatric patients based on medical evidence and implementation of the GRADE methodology (Grading of Recommendations Development and Evaluation). In addition, they identify the differential symptoms of these diseases, allowing the medical community to guide differential diagnoses in a timely manner.

This study offers a methodological structure to implement recommendation 1, taken from the most recent guidelines for diagnosing and treating dengue, Zika, and chikungunya, using a dataset featuring the symptoms of dengue and chikungunya. The aim was to transform the qualitative information of this recommendation into quantitative information in the dataset, which would allow the establishment of differential weights for the symptoms according to the medical evidence and the GRADE scale. Additionally, a comparison of different machine learning models is proposed to evaluate the results of applying this methodology. The rest of the article is organised as follows: Section 2 presents the background of the article, Section 3 analyses the methodology, Section 4 presents the results and discussion, and Section 5 concludes the paper.

## 2. Background 

### 2.1. Machine Learning in the Differential Classification of Arboviruses

Several arboviruses have similar clinical symptoms, including dengue, leptospirosis, malaria, chikungunya, and Zika [9]. Confirmatory laboratory tests are required to distinguish between these diseases, such as ELISA or RT-PCR, which detect the NS1 protein, immunoglobulin G (IgG), or immunoglobulin M (IgM), respectively, and are highly sensitive and specific [10]. However, these tests are often inaccessible in rural areas because of the need for specialised infrastructure and trained personnel, which hinders disease management. To address this challenge, researchers have proposed the use of Information and Communication Technologies (ICTs) to support arbovirus diagnosis. Among these technologies are Big Data [11,12], Artificial Intelligence [13,14], Deep Learning [15,16], and machine learning [17,18,19]. The final technology is described in detail below.

Several studies have used machine learning (ML) techniques to support medical decisions in the accurate classification of arboviruses. Most of these studies proposed binary classification models. For example, in [20,21,22,23,24], the presence of dengue or not is classified, the severity of dengue (yes or no) [25,26], the risk of dengue (yes or no) [27], the presence of chikungunya or not [28], and Zika between “discarded” and “probable” [29]. However, few studies have focused on the differential classification of arboviruses, particularly the multi-classification of dengue, Zika, and chikungunya [30].

The most recent studies on differential classification have focused on dengue and chikungunya using clinical data [31,32], but other arboviruses have also been proposed. In [33], a model based on linear and nonlinear quantum retrieval algorithms was proposed to aid in the diagnosis of malaria, yellow fever, typhoid fever, and dengue. Similarly, an RGB-based model was suggested in [34] to aid the classification of dengue, Zika, and yellow fever. In contrast, in [35], a multiclass model that used various classifiers such as SVM, KNN, MLP, and Random Forest was used to distinguish between dengue, chikungunya, and other similar diseases. Finally, in [36], it was concluded that more research is needed to focus on the timely diagnosis of arboviruses after analysing 30 methodologies for the development of ML-based algorithms.

### 2.2. Quality Metrics for Model Evaluation

These evaluation metrics serve various purposes and provide diverse measurements. This study employed the metrics of precision, recall, F1 score, and specificity described below.

#### Confusion Matrix

The confusion matrix allows for the visualisation of systematic errors in a machine learning model. The word “confusion” refers to the mislabelling of samples [30,37,38,39,40]. Figure 1 shows the structure of the two-class confusion matrix:

Several terms can be derived from the confusion matrix as follows:
positive observation.Negative (N): The observation is not positive; that is, it is negative.True Positive (TP): The model correctly predicted the positive class.True Negative (TN): The model correctly predicts the negative class.False Positive (FP): Known as a type 1 error, it occurs when the model incorrectly predicts the positive class when it is of the negative class.False Negative (FN): Known as a type 2 error, it occurs when the model incorrectly predicts the negative class when it is of the positive class.

The main metrics of the confusion matrix are as follows:

Accuracy: This metric indicates the proportion of correct predictions made by a model relative to the total predictions [39].


(1)
$$Accuraccy=\frac{TP+TN}{TP+TN+FP+FN}$$


Precision: This is known as the positive predictive value and is the ratio of relevant instances to retrieved instances [39].


(2)
$$Precision=\frac{TP}{TP+FP}$$


Sensitivity: The rate of hits or the true positive rate (TPR) was calculated as the ratio of the total number of instances retrieved. This quality metric provides an answer to the real positives that are correctly identified [40].


(3)
$$Recall=\frac{TP}{TP+FN}$$


Specificity: This metric is known as the true negative rate (TNR), and it evaluates the proportion of true negatives that are correctly identified. This is the counterpart to the sensitivity [40].


(4)
$$Precision=\frac{TN}{TN+FP}$$


F1 Score: This metric is known as the harmonic mean of precision and recall, and is a measure that takes into account all the results of the confusion matrix, allowing a metric of the precision and robustness of the model [39].


(5)
$$F1\;Score=\frac{2\times \left(precision\times recall\right)}{2TP+FP+FN}$$


### 2.3. Linear Interpolation

Linear interpolation was used to calculate the value within a given range on a straight line [41]. It is used to estimate the value between two points ($x_{0}$, $y_{0}$) and ($x_{1}$, $y_{1}$) on a straight line, determining the nearest value of *x* [42]. This method is widely used in areas that are not specialised in mathematics, such as the health and social sciences, because it can assign intermediate values within a scale [43]. The mathematical formula for linear interpolation is as follows [44,45,46]:(6)$$y=y_{0}+\frac{\left(x-x_{0})(y_{1}-y_{0}\right)}{\left(x_{1}-x_{0}\right)}$$
y=the value of the variable that you want to find.x=the value of the point that is parallel to y.$x_{0}=the\;value\;of\;the\;point\;closest\;before\;x.$$x_{1}=the\;value\;of\;the\;point\;closest\;after\;x.$$y_{0\;}=the\;value\;of\;the\;point\;closest\;before\;y.$$y_{1}=the\;value\;of\;the\;point\;closest\;after\;y.$

### 2.4. Synthesis of a Guide for the Diagnosis and Treatment of Dengue, Chikungunya, and Zika in the American Region

At the end of 2022, the PAHO published a special report titled “Evidence Synthesis: Guidelines for the Diagnosis and Treatment of Dengue, Chikungunya, and Zika in the Americas Region” [7]. This report summarises the recommendations for the proper diagnosis and treatment of these diseases. To establish the certainty of medical evidence regarding the differential symptoms of these three diseases, experts used the GRADE (Grading of Recommendations Assessment Development and Evaluation) method for rapid guideline development. Table 1 summarises the scale used by the PAHO to measure the certainty of medical evidence.

The first recommendation, relevant to this study, involves differentiating and classifying the signs and symptoms of dengue, Zika, and chikungunya, and assigning them a category according to the GRADE evidence scale mentioned earlier. Table 2 summarises this recommendation:

## 3. Materials and Methods

To develop the experiment, the latest PAHO evidence synthesis report was used, which provides guidelines for the diagnosis and treatment of dengue, chikungunya, and Zika in the American region [7]. The main challenge was to adapt the qualitative scales of certainty of evidence from the 12 recommendations in this report to a quantitative scale that would later allow them to be assigned to a dataset and create a model based on machine learning techniques for the early prediction of these diseases (Figure 2).

### 3.1. Identification of the PAHO Protocol Variables in the Dataset and Quality Rules

#### 3.1.1. Dataset Selection

Various sources were explored to select the dataset, such as Open Data Colombia, Kaggle, Mendeley Data, and Google Dataset Search, with the aim of finding datasets related to the signs and symptoms of dengue, Zika, and chikungunya. However, this process only managed to obtain data on dengue and chikungunya, including signs, symptoms, and sociodemographic information. In contrast, the information found on Zika virus infection did not include signs and symptoms; therefore, it was excluded from the study.

The selected dataset was published in Mendeley Data and is available at https://data.mendeley.com/datasets/bv26kznkjs/1 accessed on 10 February 2024. It was processed by [31]. This dataset contained data on confirmed patients with dengue and chikungunya, along with sociodemographic variables, signs, symptoms, and comorbidities from the health system in the city of Recife, Brazil, in the years 2015–2020. It contained 27 attributes, with 17,172 records evenly distributed among dengue, chikungunya, and other diseases.

Nine symptoms were identified in the dataset that were considered differential between dengue and chikungunya according to the certainty of evidence indicated in the PAHO guidelines [7], with levels of high, moderate, and low certainty. Table 3 relates each variable to its respective certainty of evidence:

Quality rules were established for age and the number of days to adjust the records in the dataset.

#### Age Rule

To establish this rule, the PAHO report on ageing was considered, which indicated that the age limit was 100 years [47]. Although it is possible that there were people older than this, this criterion was established for the purposes of this study.
(7)0 years<age≤100 years

#### Rules for the Course of Symptoms of the Disease

Regarding dengue, the WHO indicates that symptoms usually appear 4–10 days after infection [48,49], whereas for chikungunya, this period is 2–12 days [50,51]. Generally, the disease starts with a fever, and some common symptoms between the two diseases lead the infected person to seek medical attention when the record is taken. Therefore, the rules are as follows.

Rules for dengue symptoms
(8)0 days<dengue symptoms≤10 days

Rules for chikungunya symptoms
(9)0 days<Chikungunya symptoms≤12 days

### 3.2. Coding and Categorization According to the Certainty of the Evidence from the PAHO

In this phase, the scales used by the PAHO in the report [7] were identified and assigned a range of values from 0 to 1 according to the established category, as indicated in Table 4.

Table 5 shows the assignment of the quantitative weights by range, according to the category to which each symptom identified in the dataset belongs. This assignment considers the classification of diseases, symptoms, and guidelines provided in the report [7].

### 3.3. Adjusting Datast Outliers

In this initial stage, it was verified that the dataset complied with the previously established quality standards, and outliers were processed for the entire dataset. In this context, the label “Other” was excluded from the target variable, which indicated a disease different from dengue and chikungunya in the dataset. This is because the PAHO report [7] only offers recommendations for dengue, Zika, and chikungunya, and does not provide a way to assign certainty weights to the signs and symptoms of this label in the dataset.

### 3.4. Parameterise the Linear Interpolation Function

The mathematical function of linear interpolation was used to convert qualitative labels into quantitative values. In this process, the “days of illness” feature of the dataset was chosen, as it can offer a more objective and quantifiable representation of symptoms in relation to the time of their appearance. The parameterisation was performed using the following equation:(10)$$y=y_{0}+\frac{\left(x-x_{0})(y_{1}-y_{0}\right)}{\left(x_{1}-x_{0}\right)}$$
y = the interpolated value for day *x*.x = the value of the day to be interpolated.$x_{0},x_{1}\;$= the range of day values ranging from 0 to 12.$y_{0}$, $y_{1}\;$= values of the established range High, Moderate, Low, and Very Low towards which the interpolation of days is desired.

### 3.5. The Transformation from Qualitative to Quantitative Labels Was Applied Based on the Interpolation Function

The experiment proposed assigning the weight of evidence based on a mathematical relationship (linear interpolation) with the days of illness, aiming for a more meaningful correlation between the symptoms and disease progression. The descriptive variables indicating signs and symptoms in the dataset are binary, with values of “yes” or “no”, while the target variable “Target” categorizes diseases as “dengue” or “chikungunya”. To transform the symptom labels, the proposal suggests applying a pre-parameterised interpolation function and establishing specific rules using conditionals (“if” statements) from Table 3. These rules allow the assignment of evaluative weights and the replacement of the original labels. The pseudocode provided enables the replication of the experiment using any dataset containing the signs and symptoms of these diseases. Algorithm 1 shows the procedure is as follows.
**Algorithm 1** The Transformation from Qualitative to Quantitative Labels Was Applied Based on the Interpolation FunctionDefine *x*, *x*0, *x*1, *y*0, *y*1 as real numbers
Define *x*_min = 0, *x*_max = 12, cero = 0 as constantsDefine índice = 0 as an integerDefine Very_low_value_0 = 0.0, Very_low_value_1 = 0.25, Low_value_0 = 0.26, Low_value_1 = 0.50, Moderate_value_0 = 0.51, Moderate_value_1 = 0.75, High_value_0 = 0.76, High_value_1 = 1 as constantsFunction LinearInterpolation (*x*, *x*0, *x*1, *y*0, *y*1):*y = y0 + ((x − x0) * (y1 − y0)) / (x1 − x0)* Return yFor each row in the dataset dataset1: For each index and element in the row until the second-to-last element:  If the last element of the row is “chikungunya” and the current element is “yes”:   Apply the interpolation function   Assign the interpolated value to the element  Else, if the last element of the row is “chikungunya” and the current element is “no”:   Assign the value of cero to the element  Else, if the last element of the row is “dengue” and the current element is “yes”:   Apply the interpolation function   Assign the interpolated value to the element  Else, if the last element of the row is “dengue” and the current element is “no”:   Assign the value of cero to the element End forEnd for***End function***

### 3.6. Data Preprocessing

In this phase, data were prepared for effective use in the machine learning algorithms through several steps. First, irrelevant and redundant variables were eliminated to reduce the model’s complexity. Then, a statistical description was performed to understand the data distribution and identify outliers, which were addressed by correcting or removing significant deviations. Null values were managed using advanced imputation techniques. In addition, because the dataset was balanced, no further balancing was necessary to avoid bias. Finally, the variables were transformed according to the requirements of the algorithm, which may include the normalisation, standardisation, and encoding of categorical variables.

### 3.7. Hyperparameter Tuning of ML Techniques

In this stage, the hyperparameters of the selected ML techniques were configured. Initially, these hyperparameters were manually adjusted and the resulting model quality metrics were evaluated. It was not necessary to employ more advanced techniques such as grid searching for optimal hyperparameters.

### 3.8. Modelling with ML Techniques

At this stage, which is crucial for the training process, the data-splitting technique was used at a 70/30 ratio. This means that 70% of the data were allocated for model training, allowing the model to learn from these data and adjust to the patterns present in them. The remaining 30% were reserved for evaluating the performance of the model on unseen data during the training, helping to measure its generalisation ability and ability to accurately predict new data. This stratified split ensured that the classes were represented in a balanced manner in both the training and test sets, which is essential for obtaining reliable results and for avoiding overfitting.

### 3.9. Selection of the Model with the Best Result

Selecting the appropriate model is a crucial step that follows the evaluation of multiple models using metrics, such as precision, recall, F1-score, and accuracy. A confusion matrix that provides details of true positives, true negatives, false positives, and false negatives was used to calculate these metrics. The chosen model demonstrated its ability to accurately classify the test instances by outperforming these metrics.

## 4. Results and Discussion

Table 6 presents a statistical summary of the symptoms after the applied transformation. It includes the mean, standard deviation, and highest and lowest values of the transformed data, as well as the 25th, 50th, and 75th percentiles. These data confirm that the assignment made through the interpolation function, using the number of days that the symptoms were present at the time of consultation, complied with the rules established in the process, and aligned with the WHO guidelines for 2022.

Figure 3 provides a detailed visualisation of how the quantitative weights were assigned after the transformation of the arthritis variable in dengue and chikungunya. For dengue, a value of 0 was assigned for the “no” label, indicating the absence of arthritis, and a value from 0 to 0.25 was interpolated based on the days of the disease for the “yes” labels, indicating the presence of arthritis with a very low certainty of evidence. For chikungunya, the assignment was similar, with a value of 0 for the “no” label and an interpolated value based on the days of symptom presence ranging from 0.51 to 0.75. This approach to weight assignment allows for the objective quantification of arthritis based on the symptoms and days of illness, following the guidelines established by the OPS.

Table 7 presents the detailed results of the experimentation with various machine learning techniques using the transformed dataset, following the methodological proposal that assigns evaluative weights based on the guidelines established by the OPS in 2022 [7]. There was a significant balance between the four quality metrics used for the disease classification. This balance suggests that the classification model achieved a consistent and reliable performance in differentiating between dengue and chikungunya.

According to the results, it was observed that ensembles in general, such as Random Forest and Boosting, performed better than the classical techniques, such as neural networks or KNN. However, it is worth noting that decision trees showed metrics very similar to those of ensembles and, unlike these, they are more interpretable in the medical field. This characteristic makes the application of this method particularly attractive for the identification and classification of diseases, such as dengue and chikungunya.

Figure 4 shows the classification tree, highlighting that the variable “arthralgia” is the most significant for classifying the chikungunya disease. This result coincides with the OPS guidelines for 2022, which indicate a high certainty in the medical evidence that this symptom is differential in chikungunya. Likewise, the variables “myalgia”, “arthritis”, and “rash” are considered differentiating for chikungunya with a moderate certainty by the OPS, and were selected by the tree, coinciding with these guidelines.

For comparison with previous results, an experiment was carried out with the untransformed dataset, and the results are listed in Table 8.

The results presented in Table 8 confirm the balance observed in the four quality metrics, which are similar to those in Table 5. However, the performance was significantly lower than that obtained when the proposed methodology was applied.

The decision tree shown in the Figure 5 indicates that the most significant variable for classification is arthralgia, which aligns with the high certainty in the medical evidence that this symptom differs in chikungunya. Additionally, it was observed that the variables, arthritis and myalgia, are also important in the classification of this disease, with moderate evidence according to the OPS for this disease.

In contrast to the previous tree, retroocular pain was an important variable in the classification. According to the guidelines, this symptom is different from that of dengue, with a low certainty of evidence.

It is important to highlight that, both with the proposed methodology and with the dataset without transformations, the results of the classification trees align with the guidelines provided by the PAHO for the selection of differential variables between dengue and chikungunya diseases. This consistency in the results supports the validity of the proposed assignment of evaluative weights based on the PAHO guidelines for the dataset.

Furthermore, when comparing the trees generated with and without the transformation of the dataset, it can be observed that the proposed methodology achieves a more precise and effective selection of variables to differentiate diseases. This suggests that assigning evaluative weights based on the PAHO guidelines can significantly improve the quality of classification models in this medical context.

Furthermore, these results show better performance in disease classification compared with those obtained in previous studies [31,32], which also explored the same dataset. In a study [31], a precision of 70% was not achieved, and there was no balance between the quality metrics, resulting in greater difficulties in classifying dengue. 

Furthermore, these results demonstrate better performance in disease classification compared to previous studies [31,32], which can be seen in Table 9, where the same dataset was explored. In that study [31], an accuracy of 70% was not achieved, and there was an imbalance between the quality metrics, which led to greater difficulties in the classification of dengue and chikungunya. On the other hand, in [32], as in [31], multiclass classification was performed, but with the proposal of transforming the dataset into images and using a single-layer convolutional neural network (CNN), the DeepInsight CNN. However, this approach achieved a classification of less than 80%. 

It is important to clarify that the previously compared studies were the only ones that used the same dataset but performed multiclass classification. Owing to the nature of the methodological proposal, it is not possible to work with the label “others”, so the results of the comparisons may be affected by the fact that they involve binary and multiclass classifications.

These studies are important because they highlight the complexity of the data in the diagnosis of arboviruses, as they share symptoms that can be indistinguishable when classifying the diseases in question. Additionally, there is the possibility of co-infection, which represents a major challenge in clinical diagnosis.

This study aimed to assign evaluative weights based on medical evidence to overcome these challenges and improve the results of disease classification. This has led to the development of a solid methodological proposal that can be scaled to any dataset that contains these diseases.

## 5. Conclusions

Arboviruses are infections that present with similar symptoms in the early stages, making timely differential diagnosis challenging. The approaches such as those developed in this study provide valuable tools in the clinical field to support medical decisions. This tool is even more relevant in hard-to-reach areas, where specialised professionals or equipment for high-specificity tests such as PCR or IgM antibody tests are not available to diagnose patients early enough to apply specific treatment.

The methodological proposal to assign quantitative values based on recommendations supported by the certainty of medical evidence from the OPS in 2022 represents a significant advancement in the field. This methodology allows the direct application of these recommendations to datasets for the differential classification between dengue and chikungunya, achieving consistent quality metrics, and thereby contributing to the improvement in the knowledge and clinical practice in managing these diseases.

In this study, a methodology that assigns quantitative values to the symptoms of dengue and chikungunya based on the certainty of medical evidence from the OPS using interpolation techniques was proposed. This methodology has improved the quality of machine learning models for classifying diseases by providing more accurate evaluative weights. These findings imply that this approach can significantly enhance the detection and management of these diseases.

According to the results obtained, ensemble machine learning methods, such as Random Forest and Boosting, outperform traditional techniques, such as neural networks or KNN. However, it is important to recognise that decision trees exhibit metrics that are highly comparable to those of ensembles and that they are simpler to interpret in a medical context than ensembles, which makes them particularly appealing for the identification and classification of diseases such as dengue and chikungunya. Furthermore, the analysis of the classification tree reveals the importance of variables such as “arthralgia”, “myalgia”, “arthritis”, and “rash” in the differentiation of chikungunya, in line with the OPS guidelines that support the relevance of these symptoms in differential diagnosis.

In future work, we plan to apply the methodology developed in this study to a dataset containing all three diseases—dengue, Zika, and chikungunya—as established in the OPS 2022 recommendations, which share similar symptoms at disease onset. The early differential classification of these three diseases in endemic areas represents a significant challenge; therefore, a clinical support model that covers all three diseases is of great importance to the medical community. Furthermore, medical validation should be conducted in the clinical setting to assess the efficiency of this model for disease classification.

## Figures and Tables

**Figure 1 viruses-16-01088-f001:**
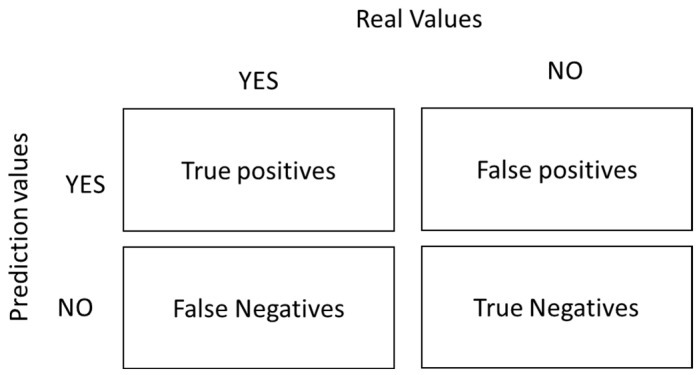
Structure of the confusion matrix.

**Figure 2 viruses-16-01088-f002:**
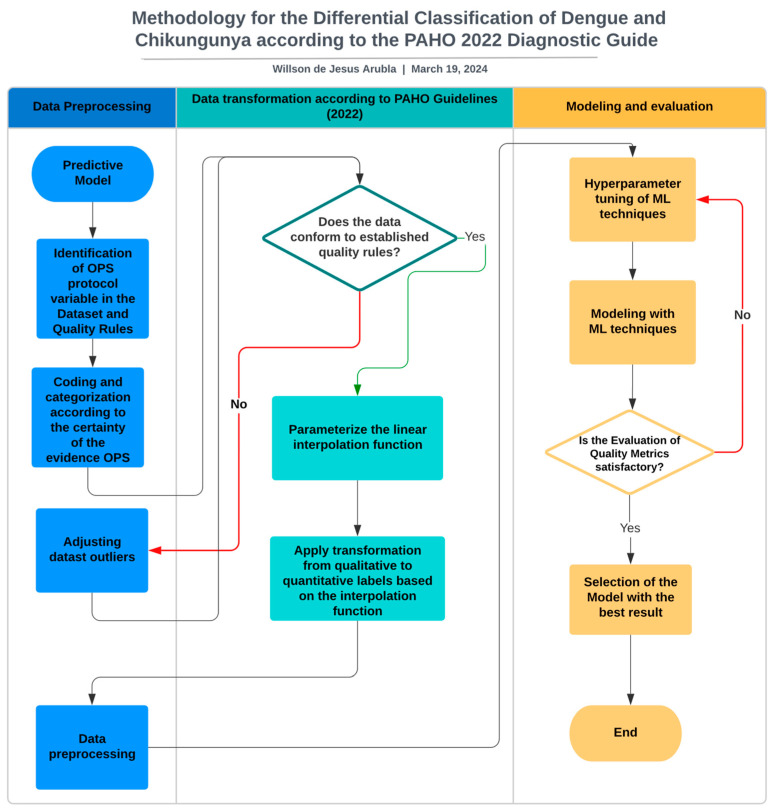
Flowchart of the methodological proposal for the development of a predictive model for dengue and chikungunya based on OPS diagnostic guidelines.

**Figure 3 viruses-16-01088-f003:**
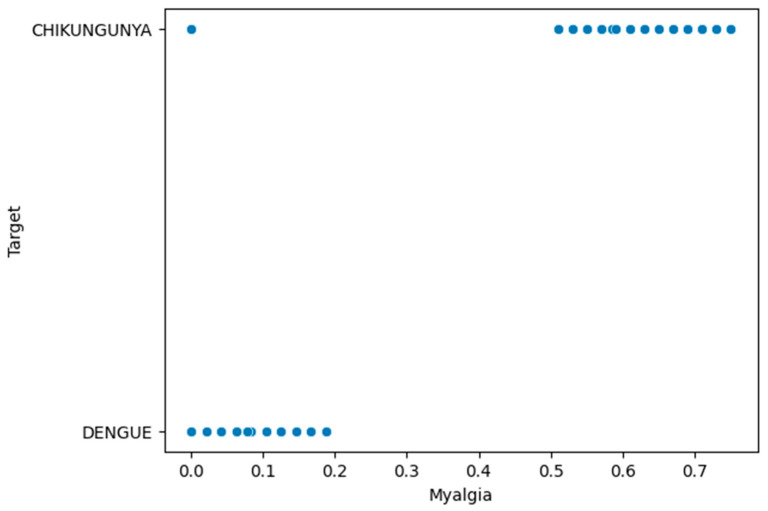
Quantitative transformation of qualitative labels for arthritis.

**Figure 4 viruses-16-01088-f004:**
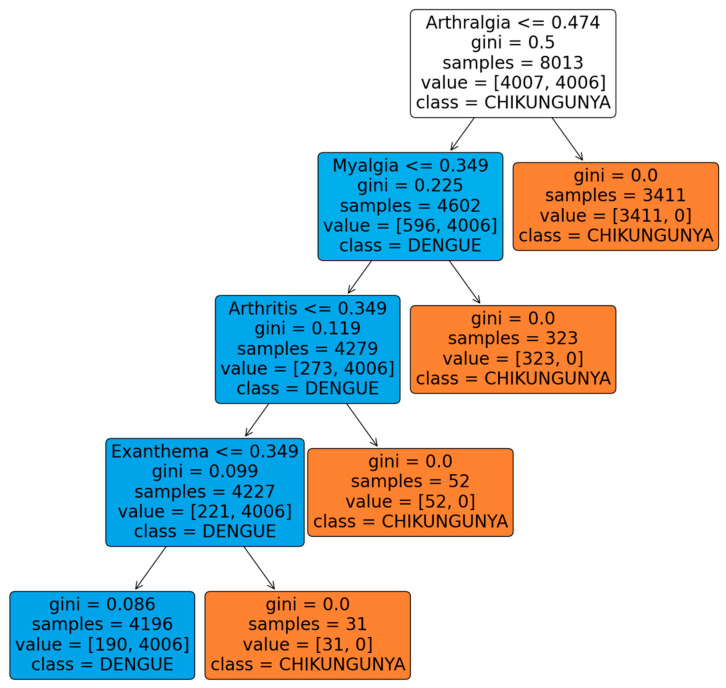
Dengue and chikungunya disease classification tree using evaluative weights based on PAHO 2022 guidelines.

**Figure 5 viruses-16-01088-f005:**
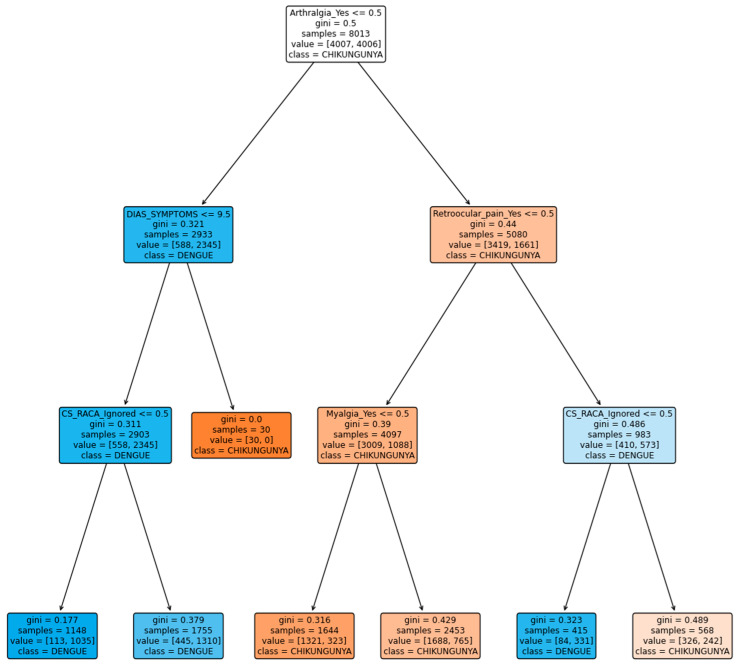
Classification tree for dengue and chikungunya diseases without using evaluative weights based on the OPS 2022 guidelines.

**Table 1 viruses-16-01088-t001:** Certainty of evidence according to the GRADE system [7].

Certainty in the Evidence According to the PAHO Guidelines (2022)	Meaning
High    	Further studies are unlikely to change the confidence in the estimated result.
Moderate    	New studies could have a significant impact on the confidence in the result.
Low    	New studies have a high probability of significantly impacting the confidence in the estimated result, potentially modifying it.
Very Low    	The level of certainty regarding any estimated results is very low.

Note: 

 equals 25% and 

 0%.

**Table 2 viruses-16-01088-t002:** Evidence for signs and symptoms of dengue, Zika, and chikungunya [7].

Certainty in the Evidence According to the PAHO Guidelines (2022)	Manifestations of Dengue	Manifestations of Chikunguña	Manifestations of Zika
High    	ThrombocytopeniaProgressive increase ofhaematocritLeukopenia	Arthralgias	Pruritus
Moderate    	Anorexia or hyporexiaVomitingAbdominal painShaking chillsHaemorrhages (includesbleeding on the skin,mucous membranes or both)	RashConjunctivitisArthritisMyalgia or bone pain	RashConjunctivitis
Low    	Retroocular painHepatomegalyHeadacheDiarrhoeaDysgeusiaCoughElevation oftransaminasesTourniquet testpositive	Bleeding (includes bleedingon the skin or mucous membranes)	LymphadenopathyPharyngitis/odynophagia
Very Low    	−	−	−

Note: 

 equals 25% and 

 0%.

**Table 3 viruses-16-01088-t003:** Differential variables according to the certainty of evidence from the PAHO [7].

Variable	Certainty in the Evidence According to the PAHO Guidelines (2022)	
	Demonstrations in Dengue	Demonstrations in Chikungunya
Myalgia	−	Moderate
Headache	Low	−
Exanthema	−	Moderate
Threw up	Low	−
Conjunctivitis	−	Moderate
Arthritis	−	Moderate
Arthralgia	−	High
Laco (symptom—tourniquet test)	Low	−
Retroocular pain	Low	−

**Table 4 viruses-16-01088-t004:** Assignment of quantitative weights to categories proposed by the PAHO guidelines (2022).

Certainty in the Evidence According to the PAHO Guidelines (2022)	Meaning	Quantitative Value
High    	Further studies are unlikely to change the confidence in the estimated result.	0.76–1
Moderate    	New studies could have a significant impact on the confidence in the result.	0.51–0.75
Low    	New studies have a high probability of significantly impacting the confidence in the estimated result, potentially modifying it.	0.26–0.50
Very Low    	The level of certainty regarding any estimated results is very low.	0–0.25

Note: 

 equals 25% and 

 0%.

**Table 5 viruses-16-01088-t005:** Assignment of weights to variables identified in the dataset according to the certainty of the PAHO evidence.

Variable	Certainty in Evidence		Quantitative Weight Assignment
	Manifestations in Dengue	Demonstrations in Chikungunya	
Myalgia	*	Moderate	0.51–0.75
Headache	Low	*	0.26–0.50
Exanthema	*	Moderate	0.51–0.75
Threw up	Low	*	0.26–0.50
Conjunctivitis	*	Moderate	0.51–0.75
Arthritis	*	Moderate	0.51–0.75
Arthralgia	*	High	0.76–1
Laco (symptom—tourniquet test)	Low	*	0.26–0.50
Retroocular pain	Low	*	0.26–0.50

Note: * A quantitative weight of very low certainty of evidence (0.0–0.25) was assigned when the label “yes” was present, because any result is uncertain according to the certainty of evidence in the GRADE system.

**Table 6 viruses-16-01088-t006:** Statistics of the symptoms after transformation.

	Myalgia	Headache	Exanthema	Threw Up	Conjunctivitis	Arthritis	Arthralgia	Laco (Symptom—Tourniquet Test)	Retroocular Pain
count	11,448	11,448	11,448	11,448	11,448	11,448	11,448	11,448	11,448
mean	0.199	0.140	0.098	0.081	0.014	0.041	0.373	0.006	0.045
std	0.255	0.151	0.206	0.195	0.083	0.143	0.403	0.044	0.110
min	0	0	0	0	0	0	0	0	0
25%	0	0	0	0	0	0	0	0	0
50%	0.078	0.078	0	0	0	0	0.078	0	0
75%	0.53	0.32	0.078	0	0	0	0.835	0	0
max	0.75	0.44	0.75	0.69	0.75	1	0.44	0.44	0.44

**Table 7 viruses-16-01088-t007:** Results of the experimentation with ML techniques using evaluative weights based on the OPS 2022 guidelines.

ML Technique	Accuracy	Precision	Recall	F1-Score
Tree Decision	98.5%	99%	99%	99%
KNN	81%	81%	81%	81%
Neural Network	98%	98%	98%	98%
SVM	98%	97%	97%	97%
RF	99%	99%	99%	99%
Baggin	98%	98%	98%	98%
Boosting	98%	98%	98%	98%
Hard-voting	99%	99%	99%	99%
Soft-voting	98%	98%	98%	98%
Stacking	99%	99%	99%	99%

**Table 8 viruses-16-01088-t008:** Results of the experimentation with ML techniques without using evaluative weights based on the OPS 2022 guidelines.

ML Technique	Accuracy	Precision	Recall	F1-Score
Tree Decision	75%	75%	75%	75%
KNN	68%	68%	68%	68%
Neural Network	77%	77%	77%	77%
SVM	73%	73%	73%	73%
RF	78%	79%	78%	78%
Baggin	77%	77%	77%	77%
Boosting	71%	71%	71%	71%
Hard-voting	79%	79%	79%	79%
Soft-voting	77%	78%	77%	77%
Stacking	79%	79%	79%	79%

**Table 9 viruses-16-01088-t009:** Comparison of the results with other similar studies that used the same dataset.

Author	ML Technique	Accuracy	Precision	Recall	F1-Score
Proposed methodology	Tree Decision	98.5%	99%	99%	99%
[31]	GBM	62.4%	62.5%	62%	61.9%
[32]	Tune Deepinsight CNN	75%	74.8%	74%	74%

## Data Availability

The data presented in this study are available upon request from the corresponding author.

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
