# Peer review of "Methodology for the Differential Classification of Dengue and Chikungunya According to the PAHO 2022 Diagnostic Guide"

_viruses, 2024, doi:10.3390/v16071088_

Round 1
Reviewer 1 Report
Comments and Suggestions for Authors
The evaluated article clearly presents its objectives and results. It is pertinent to the field of research discussed and is well-structured in its writing.
A significant number of the cited references are from the last five years and are relevant to the research field. Some older references pertain to international policy documents that have not been updated in recent years. Other references, which are 10 or more years old, reflect the limited literature available in the Latin American context within the research area. I found two self-citations, indicating that the authors have previously worked on similar or related topics.
The article contains 5 figures and 8 tables that are easy to understand and interpret, and they are relevant for illustrating the method used to obtain the results. For the research, a public dataset from the health system of Recife (Brazil) was used, containing 17,172 records with 27 different attributes of data collected between 2015 and 2020.
The authors conclude that assigning quantitative values to the symptoms of dengue and chikungunya based on medical evidence from the Pan American Health Organization (PAHO) and using interpolation techniques enhances the quality of machine learning models for disease classification by providing more accurate evaluative weights. Ensemble methods (Random Forest and Boosting) are more precise than traditional techniques (neural networks, KNN). However, it is acknowledged that decision tree-based techniques are comparable to ensemble methods and are easier to understand for the identification and classification of diseases such as chikungunya and dengue. These results are consistent with the methodological development presented in the article.
The dataset used in the study is publicly available, downloaded from the Mendeley database.
The literature review is clear and relevant. It explores 37 different sources to establish the knowledge gap that the research aims to address: improving the early diagnosis of diseases such as dengue and chikungunya according to the PAHO guidelines published in 2022.
The research is rigorous regarding the methodology employed to achieve its results. As the article proposes a methodology for improving the differential and timely diagnosis of diseases such as dengue, zika, and chikungunya, the authors could have formulated a hypothesis related to their research work.
The article details the process through which the proposed methodology for diagnosing dengue and chikungunya was developed. These results are applicable to a dataset similar to the one used during the investigation.
The conclusions are coherent and supported by the data presented in the results. No citations are used to support the reported conclusions.
The presented results align with the scope of the journal.
Author Response
Dear reviewer,
We would like to thank you for the review and the valuable comments provided for our article titled “Methodology for the Differential Classification of Dengue and Chikungunya according to the PAHO 2022 Diagnostic Guide.” We have worked diligently to address your observations and suggestions, and I am pleased to inform you that we have completed the requested corrections.
Below, we have summarised the main modifications made in response to your comments.
Observation reviewer :
- The evaluated article clearly presents its objectives and results. It is pertinent to the field of research discussed and is well-structured in its writing.
A significant number of the cited references are from the last five years and are relevant to the research field. Some older references pertain to international policy documents that have not been updated in recent years. Other references, which are 10 or more years old, reflect the limited literature available in the Latin American context within the research area. I found two self-citations, indicating that the authors have previously worked on similar or related topics.
The article contains 5 figures and 8 tables that are easy to understand and interpret, and they are relevant for illustrating the method used to obtain the results. For the research, a public dataset from the health system of Recife (Brazil) was used, containing 17,172 records with 27 different attributes of data collected between 2015 and 2020.
The authors conclude that assigning quantitative values to the symptoms of dengue and chikungunya based on medical evidence from the Pan American Health Organization (PAHO) and using interpolation techniques enhances the quality of machine learning models for disease classification by providing more accurate evaluative weights. Ensemble methods (Random Forest and Boosting) are more precise than traditional techniques (neural networks, KNN). However, it is acknowledged that decision tree-based techniques are comparable to ensemble methods and are easier to understand for the identification and classification of diseases such as chikungunya and dengue. These results are consistent with the methodological development presented in the article.
The dataset used in the study is publicly available, downloaded from the Mendeley database.
The literature review is clear and relevant. It explores 37 different sources to establish the knowledge gap that the research aims to address: improving the early diagnosis of diseases such as dengue and chikungunya according to the PAHO guidelines published in 2022.
The research is rigorous regarding the methodology employed to achieve its results. As the article proposes a methodology for improving the differential and timely diagnosis of diseases such as dengue, zika, and chikungunya, the authors could have formulated a hypothesis related to their research work.
The article details the process through which the proposed methodology for diagnosing dengue and chikungunya was developed. These results are applicable to a dataset similar to the one used during the investigation.
The conclusions are coherent and supported by the data presented in the results. No citations are used to support the reported conclusions.
The presented results align with the scope of the journal.
Response: According to the comments, the suggestion can only be evidenced with two self-citations, so it was decided to eliminate one of the citations number 4. The reviewer indicates this from the perspective that work is being done on the topic.
- reference removed:
- Arrubla, W.D.J.A. Conceptualización del diagnóstico del Dengue desde una perspectiva de la ingeniería y las nuevas tecnologías. Computer and Electronic Sciences: Theory and Applications 2022, 3, 1–8, doi:10.17981/cesta.03.01.2022.01
We remain at your disposal to provide any additional information or clarification that you may need. We have attached a revised copy of the manuscript along with this letter for your final review. If you find everything to be in order, we would appreciate your confirmation before proceeding with the publication.
Thank you again for your consideration, and look forward to hearing from you soon.

Reviewer 2 Report
Comments and Suggestions for Authors In chapter 2 the numbering should be reviewedDo not go into such detail in the proposed metrics
Be more forceful in data processing
More M.L techniques can be used that allow comparing results with those of the state of the art
Author Response
Dear reviewer,
We would like to thank you for the review and the valuable comments provided for our article titled “Methodology for the Differential Classification of Dengue and Chikungunya according to the PAHO 2022 Diagnostic Guide.” We have worked diligently to address your observations and suggestions, and I am pleased to inform you that we have completed the requested corrections.
Below, we have summarised the main modifications made in response to your comments.
- Observation reviewer: In chapter 2 the numbering should be reviewed
Response:
Before: 3.4. Synthesis of a guide for the diagnosis and treatment of dengue, chikungunya, and Zika in the American Region.
Now (see line 144 to 145)
2.4. Synthesis of a guide for the diagnosis and treatment of dengue, chikungunya, and Zika in the American region.
- Observation reviewer: Do not go into such detail in the proposed metrics.
Response:
Before
Positive (P): The observation is positive (e.g. it is a cat).
Negative (N): The observation is not positive, that is, it is negative (e.g. it is not a cat).
Now (see line 107 to 108)
Positive (P): positive observation.
Negative (N): The observation is not positive, that is, it is negative
- Observation reviewer: Be more forceful in data processing
Response:
Before:
3.6. Data preprocessing
In this phase, the data are prepared for effective use using machine learning algorithms. The following steps outline the dataset.
3.6.1. Elimination of irrelevant and redundant variables
The aim is to reduce the model's complexity and improve its efficiency by discarding information that does not contribute to the analysis or that is duplicated in the other variables.
3.6.2. Statistical Description of the Data
An analysis was performed to understand the data distribution, identify outliers, and obtain a general overview of the data characteristics.
3.6.3. Treatment of outliers
They are reviewed again to identify outliers and to correct or remove values that deviate significantly from most of the data because they can negatively affect the accuracy of the model.
3.6.4. Handling of null values
The approach to handling missing values in a dataset is decided either by removing them, replacing them with a specific value, or using more advanced imputation techniques.
3.6.5. Data Balancing
Although the dataset was balanced, if it was unbalanced, data balancing was performed at this stage to equalise the distribution and avoid biases in the model.
3.6.6. Variable transformation
The variables are modified to suit the needs of the chosen machine learning algorithm, which may involve normalising or standardising the data, as well as converting categorical variables into numerical variables through appropriate encoding.
Now (see line 249 to 258)
3.6. Data preprocessing
In this phase, data are prepared for effective use in machine-learning algorithms through several steps. First, irrelevant and redundant variables were eliminated to reduce model complexity. Then, a statistical description was performed to understand the data distribution and identify outliers, which were addressed by correcting or removing significant deviations. Null values were managed using advanced imputation techniques. In addition, because the dataset was balanced, no further balancing was necessary to avoid bias. Finally, variables are transformed according to the requirements of the algorithm, which may include the normalisation, standardisation, and encoding of categorical variables.
Observation reviewer: More M.L techniques can be used that allow comparing results with those of the state of the art
Response:
Before:
Table 7. Results of experimentation with ML techniques using evaluative weights based on OPS 2022 guidelines.
|
ML technique |
accuracy |
precision |
recall |
F1- Score |
|
Tree Decision |
98.5% |
99% |
99% |
99% |
|
Knn |
81% |
81% |
81% |
81% |
|
Neural Network |
98% |
98% |
98% |
98% |
|
RF |
99% |
99% |
99% |
99% |
|
Baggin |
98% |
98% |
98% |
98% |
|
Boosting |
98% |
98% |
98% |
98% |
|
Hard-Voting |
99% |
99% |
99% |
99% |
|
Soft- voting |
98% |
98% |
98% |
98% |
|
Stacking |
99% |
99% |
99% |
99% |
Table 8. Results of experimentation with ML techniques without using evaluative weights based on OPS 2022 guidelines.
|
ML technique |
accuracy |
precision |
recall |
F1- Score |
|
Tree Decision |
75% |
75% |
75% |
75% |
|
Knn |
68% |
68% |
68% |
68% |
|
Neural Network |
77% |
77% |
77% |
77% |
|
RF |
78% |
79% |
78% |
78% |
|
Baggin |
77% |
77% |
77% |
77% |
|
Boosting |
71% |
71% |
71% |
71% |
|
Hard-Voting |
79% |
79% |
79% |
79% |
|
Soft- voting |
77% |
78% |
77% |
77% |
|
Stacking |
79% |
79% |
79% |
79% |
Now (see line 304 and 322)
Table 7. Results of experimentation with ML techniques using evaluative weights based on the OPS 2022 guidelines.
|
ML technique |
accuracy |
precision |
recall |
F1- Score |
|
Tree Decision |
98.5% |
99% |
99% |
99% |
|
Knn |
81% |
81% |
81% |
81% |
|
Neural Network |
98% |
98% |
98% |
98% |
|
SVM |
98% |
97% |
97% |
97% |
|
RF |
99% |
99% |
99% |
99% |
|
Baggin |
98% |
98% |
98% |
98% |
|
Boosting |
98% |
98% |
98% |
98% |
|
Hard-Voting |
99% |
99% |
99% |
99% |
|
Soft- voting |
98% |
98% |
98% |
98% |
|
Stacking |
99% |
99% |
99% |
99% |
Table 8. Results of experimentation with ML techniques without using evaluative weights based on OPS 2022 guidelines.
|
ML technique |
accuracy |
precision |
recall |
F1- Score |
|
Tree Decision |
75% |
75% |
75% |
75% |
|
Knn |
68% |
68% |
68% |
68% |
|
Neural Network |
77% |
77% |
77% |
77% |
|
SVM |
73% |
73% |
73% |
73% |
|
RF |
78% |
79% |
78% |
78% |
|
Baggin |
77% |
77% |
77% |
77% |
|
Boosting |
71% |
71% |
71% |
71% |
|
Hard-Voting |
79% |
79% |
79% |
79% |
|
Soft- voting |
77% |
78% |
77% |
77% |
|
Stacking |
79% |
79% |
79% |
79% |
We remain at your disposal to provide any additional information or clarification that you may need. We have attached a revised copy of the manuscript along with this letter for your final review. If you find everything to be in order, we would appreciate your confirmation before proceeding with the publication.
Thank you again for your consideration and look forward to hearing from you soon.

Reviewer 3 Report
Comments and Suggestions for Authors
The methodology is based on a review of the medical literature and content validation. This ensures that the weights assigned to symptoms are supported by solid scientific evidence. In addition, by applying this methodology, a more accurate classification of arboviruses is achieved, which is essential for early diagnosis and appropriate treatment of these diseases.
On the other hand, the proposal addresses some gaps in the field of study by following the latest guidelines provided by WHO. Although initially applied only to a dengue and chikungunya dataset, it is possible to extend it to all three diseases: dengue, zika and chikungunya, which share a similar clinical picture at an early stage of the disease. Since there are few studies focused on these three diseases, this research is relevant.
In the results, the tree algorithm shows good performance according to the proposed results, allowing the medical community to understand the predictions made.
This proposal has the potential to significantly improve medical care and epidemic management.
Recommendations for acceptance:
- Correct in section 2 (Background), the sequence of item: 3.4. Synthesis of a guide for the diagnosis and treatment of dengue, chikungunya, and Zika in the American Region. The correct sequence should be: 2.4
- Correct in section 3 (Materials and Methods), item 3.1, is currently in Spanish, it should be translated to English.
- Correct the paragraph of item 3.5. It should present in a clearer way the idea that is tried to be transmitted.
- If possible, make a comparative table according to the literature if there are other machine learning models made with the same dataset used in your experimentation to take them into account in the discussion.
Author Response
Dear reviewer,
We would like to thank you for the review and the valuable comments provided for our article titled “Methodology for the Differential Classification of Dengue and Chikungunya according to the PAHO 2022 Diagnostic Guide.” We have worked diligently to address your observations and suggestions, and I am pleased to inform you that we have completed the requested corrections.
Below, we have summarised the main modifications made in response to your comments.
Observation reviewer: Recommendations for acceptance
- Correct in section 2 (Background), the sequence of item: 3.4. Synthesis of a guide for the diagnosis and treatment of dengue, chikungunya, and Zika in the American Region. The correct sequence should be: 2.4
Response:
Before: 3.4. Synthesis of a guide for the diagnosis and treatment of dengue, chikungunya, and Zika in the American Region.
Now (see line 144 and 145)
2.4. Synthesis of a guide for the diagnosis and treatment of dengue, chikungunya, and Zika in the American region.
- Correct in section 3 (Materials and Methods), item 3.1, is currently in Spanish, it should be translated to English.
Response:
Before: 3.1. Identificación de la variable de protocolo de la OPS en el conjunto de datos y reglas de calidad
Now (see line 172)
3.1. Identification of the PAHO protocol variables in the dataset and quality rules.
- Correct the paragraph of item 3.5. It should present in a clearer way the idea that is tried to be transmitted.
Response
Before:
3.5. The transformation from qualitative to quantitative labels was applied based on the interpolation function.
In the experiment, it was proposed that the weight of the evidence be assigned based on a mathematical relationship (linear interpolation) of the days of illness. This allows a more meaningful relationship to be established between the symptoms and the course of the disease over time.
The descriptive variables that relate to the signs and symptoms in the dataset have two possible values, "yes" and "no,” while the target variable "Target" contains the labels of the disease’s "dengue" and "chikungunya.” To transform the labels of the symptoms, it is proposed to apply the previously parameterised interpolation function and establish specific rules with conditionals "if”, derived from Table 3, which allow for assigning evaluative weights and replacing said labels. The pseudocode enables replication of the experiment with any dataset that includes the signs and symptoms of these diseases. The algorithm that allows the procedure to be performed is as follows:
Now (see line 236 and 248)
3.5. The transformation from qualitative to quantitative labels was applied based on the interpolation function.
The experiment proposed assigning the weight of evidence based on a mathematical relationship (linear interpolation) with the days of illness, aiming for a more meaningful correlation between the symptoms and disease progression. Descriptive variables indicating signs and symptoms in the dataset are binary, with values of "yes" or "no," while the target variable "Target" categorizes diseases as "dengue" or "chikungunya." To transform the symptom labels, the proposal suggests applying a pre-parameterised interpolation function and establishing specific rules using conditionals ("if" statements) from Table 3. These rules allow the assignment of evaluative weights and replacement of the original labels. The pseudocode provided enables replication of the experiment using any dataset containing the signs and symptoms of these diseases. The algorithm for this procedure is as follows.
- If possible, make a comparative table according to the literature if there are other machine learning models made with the same dataset used in your experimentation to take them into account in the discussion.
Response:
Before:
Furthermore, these results show better performance in disease classification compared to those obtained in previous studies [32] and [33], which also explored the same dataset. In study [31], a precision of 70% was not achieved, and there was no balance between the quality metrics, resulting in greater difficulties in classifying dengue. This may be attributed to the binary-class approach proposed in this research, as opposed to the multiclass approach used by the authors of [32], who kept the "other" label in the dataset. The decision to remove this label in our study allowed the application of the methodological proposal of assigning weights based on the PAHO guidelines, which likely contributed to improving the classification results.
On the other hand, in [33], as in [32], multiclass classification was performed, but with the proposal of transforming the dataset into images and using single-layer convolutional neural networks (CNN) DeepInsight CNN. However, this approach achieved a classification of less than 80%. These studies are important because they highlight the complexity of the data in the diagnosis of arboviruses as they share symptoms that can be indistinguishable when classifying the diseases in question. Additionally, there is a possibility of co-infection, which represents a major challenge in clinical diagnosis.
Now (see line 351 and 374)
Furthermore, these results demonstrate better performance in disease classification compared to previous studies [31,32], which can be seen in Table 9, where the same dataset was explored. In that study [31], an accuracy of 70% was not achieved, and there was an imbalance between the quality metrics, which led to greater difficulties in the classification of dengue and chikungunya. On the other hand, in [32], as in [31], multiclass classification was performed, but with the proposal of transforming the dataset into images and using single-layer convolutional neural network (CNN) DeepInsight CNN. However, this approach achieved a classification of less than 80%.
It is important to clarify that the previously compared studies were the only ones that used the same dataset but performed multiclass classification. Owing to the nature of the methodological proposal, it is not possible to work with the label "others”, so the results of the comparisons may be affected by the fact that they involve binary and multiclass classifications.
These studies are important because they highlight the complexity of the data in the diagnosis of arboviruses, as they share symptoms that can be indistinguishable when classifying the diseases in question. Additionally, there is the possibility of co-infection, which represents a major challenge in clinical diagnosis.
This study aims to assign evaluative weights based on medical evidence to overcome these challenges and improve the results of disease classification. This has led to the development of a solid methodological proposal that can be scaled to any dataset that contains these diseases.
We remain at your disposal to provide any additional information or clarification that you may need. We have attached a revised copy of the manuscript along with this letter for your final review. If you find everything to be in order, we would appreciate your confirmation before proceeding with the publication.
Thank you again for your consideration and look forward to hearing from you soon.
